# Numerical Analysis of the Creep and Shrinkage Experienced in the Sydney Opera House and the Rise of Digital Twin as Future Monitoring Technology

**Faham Tahmasebinia [1,2,\*]**, **Daniel Fogerty [2]**, **Lang Oliver Wu [2]**, **Zhichao Li [2]**,
**Saleh Mohammad Ebrahimzadeh Sepasgozar [3]**, **Kai Zhang [2]**, **Samad Sepasgozar [1]** **and**
**Fernando Alonso Marroquin [2]**

1   Faculty of Built Environment, The University of New South Wales, Sydney, NSW 2052, Australia;
    Samad.sepasgozar@gmail.com
2   School of Civil Engineering, The University of Sydney, Sydney, NSW 2006, Australia;
    dfog3763@uni.sydney.edu.au (D.F.); lawu2488@uni.sydney.edu.au (L.O.W.);
    zhli8583@uni.sydney.edu.au (Z.L.); kzha7676@uni.sydney.edu.au (K.Z.);
    fernando.alonso@sydney.edu.au (F.A.M.)
3   Babol Noshirvani University of Technology, Babol 47148, Iran; s.sepasgozar@yahoo.com
\*   Correspondence: F.tahmasebinia@unsw.edu.au; Tel.: +61-02-9351-2171

**Abstract:** This paper presents a preliminary finite element model in Strand7 software to analyse creep and shrinkage effects on the prestressed concrete ribs of the Sydney Opera House as remarkable heritage. A linear static analysis was performed to investigate the instantaneous impacts of dead and wind loads on the complex concrete structure which was completed in 1973. A quasistatic analysis was performed to predict the effects of creep and shrinkage due to dead load on the structure in 2050 to discern its longevity. In 2050, the Sydney Opera House is expected to experience 0.090% element strain due to creep and shrinkage and therefore suffer prestress losses of 32.59 kN per strand. However, given that the current time after prestress loading is approximately 50 years, the majority of creep and shrinkage effects have already taken place with 0.088% strain and 32.12 kN of prestress losses. The analysis concludes that very minor structural impacts are expected over the next 30 years due to creep and shrinkage, suggesting a change in conservation focus from large structural concerns to inspection and maintenance of minor issues of surface cracking and water ingress. The analysis is the first step in the application of more complex finite element modelling of the structure with the integration of complex building information models. The main motivation to undertake the current numerical simulation is to determine a cost-effective solution when it comes to the long-term time-dependent analysis. The paper also will suggest future directions for monitoring unique historical buildings, including 'digital twin'.

**Keywords:** creep; shrinkage; numerical modelling; Sydney Opera House; digital twin; digital model; maintenance; monitoring sensors; building information model (BIM); heritage; historical building

## 1. Introduction

Heritage buildings contribute to cultural identity for future generations, and to economies by supporting the tourism industry. This motivates governments to invest in designing remarkable buildings for the future and identify new technologies to record, analyse, monitor and protect valuable historical buildings in the long term. In 1956, an international competition was publicised by Joe Cahill, Premier of New South Wales, for the design of an opera house in the heart of Sydney embracing Sydney's harbour at Bennelong Point. Danish architect Jorn Utzon won the contest with his unique

shell design, imaginatively fitting the structure naturally into the harbour setting as though "sails of its yaughts [1].

The Sydney Opera House was to be constructed by the Danish engineering firm Ove Arup and Partners and Australian building contractors M R Hornibrook at a budget of A$18.4 million. However, due to the complexity of its design and change in key project executives over the 16-year construction period (1957–1973), the cost blew out to around A$102 million. The structure opened in 1973, with its completion attributed to the use of early computers for structural design calculations [2]. It was deemed an instant success, becoming a national icon and symbol of Australia to the world, recognised with its 2007 inscription into the UNESCO World Heritage List and its estimated four million visitors each year. The structure includes five large performance auditoria, as well as the reception hall, rehearsal studios, backstage dressing rooms, bars, restaurants and a library [1]. Further details of the structure are given in Table 1.

Given the performance nature of the Sydney Opera House's functionality, regular modifications are made to the building's interior, creating the need for data tracking of the 'as built' design of the structure. With development of new surveying and data logging technologies, a team led by Arup have begun the progressive task of constructing a building information model (BIM) for the management of contracts and documentation for works on the site. The model, as described by the Cooperative Research Centre for Construction Innovation [3], aims to revolutionise the facilities management of the Opera House by providing contractors with all of the relevant information on the area of the structure requiring works during project tender and requiring submission of survey data for the same location post works for continued development of the model. Modelling the complex structure generated interest with the potential of new technologies in ongoing structural assessment ([3–5]).

Due to the Sydney Opera House's heritage significance and the complexity of the prestressed concrete in its design, regular inspection and maintenance of its 2194 concrete sections and over 350 km of pretensioned steel cables is required. In 2014, research into an automated system for the inspection of structural elements was initiated by the University of Sydney in collaboration with Arup, funded by the Getty Foundation's Keeping it Modern grant program [6] demonstrate the cooperation of this research with the implementation of Building Information Modelling of the Opera House to catalogue each section of its complex concrete structure, with the intention of producing an element-specific, structural conservation management plan. This paper outlines the creation of a finite element model in Strand7 software for the analysis of creep and shrinkage of the pretensioned structural concrete elements of the Sydney Opera House West Wing under the impact of dead loads. Quasistatic analysis provides a preliminary insight into the impacts on the Sydney Opera House leading up to 2050 and beyond, showing the structure's design future for the development of potential conservation solutions and the requirement to include building information modelling in finite element analysis software.

**Table 1.** Building details of Sydney Opera House [6].

| | |
|---:|:---|
| **Location:** Sydney, Australia | **Opening year:** 1973 |
| **Architects:** Jorn Utzon, Peter Hall | **Approximate cost:** $102 million AUD |
| **Structural engineers:** Ove Arup & Partners | **Height:** 65 m |
| **Function:** Performance facility | **Capacity:** 5532 patrons |

## 2. Finite Element Modelling Strategy

### 2.1. Shell

The precast, prestressed concrete ribs that make up the structure of the Sydney Opera House's shell form were developed to ensure that all of the shells hold the same curved shape. This was achieved by envisaging each shell as a segment of a much larger 75 metre radius sphere [7]. However, Jorn Utzon's Red Book [8] shows that these ribs do not share a uniform cross-section, as each segment incorporates a varied thickness and shape dependent on their location within the structure.

Post-tensioned cables were pulled through the precast concrete ribs laterally and longitudinally to restrain the segments of the structure and place the concrete cross-sections under compressive stress. The cables were initially stressed to 159 kN before grouting (University of Sydney, 2016).

Each shell component of the various wings of the structure was held to the adjacent shells using ridge elements for connection of the two sides of the shells and precast concrete semi-arches connecting shells along the structure. The semi-arches are shaped in the opposite direction to the main shells and feed back into the main shells to connect deep in its shape. On the back side of the semi-arch, a large raised precast concrete segment is used to connect to the back of a smaller main shell. Due to lack of information about the ridge piece, it was not included in the Strand7 model, but the semi-arches and precast segments are scaled from Jorn Utzon's [9] Yellow Book for use in the model.

The signature white tiling that gives the Sydney Opera House its glistening finish is laid atop lids that separate the tiles from the main concrete structure. Around 4000 lids and 1,400,000 tiles are perched on top of the Sydney Opera House [10]. As a result of the excessive detail involved in adding tiles and lids to the design, the model was simplified by excluding these elements. The loss of model weight caused by exclusion of the tiles and lids is addressed using solid concrete rib sections rather than the hollow sections found in the structure. This model simplification is also necessary for constructability and compatibility of the Strand7 model.

### 2.2. Interior

In fabricating the most versatile acoustic environment for the structure's performance halls, a lightweight, easily replaceable or modifiable design was required to produce the best audible experience for operagoers. Due to the requirement for this aspect of the building, it was constructed as a non-structural element [8] and is hence not included in the Strand7 model. The choice to use plywood timber in the interior design centred on the flexibility and lightweight nature of the timber and the strength with which it can hold the deformed shapes required for optimal acoustic reflection [7].

### 2.3. Windows

Jorn Utzon's [8] design of the Sydney Opera House calls for immense glass panelling to seal the ends of the shell structures, while allowing natural light to enter the space. As a result of the general fragile nature of glass, this aspect of the building is non-load bearing and thus not included in the Strand7 model. The glass is laminated to increase the impact strength of the panels and is supported by steel framing, which is also not included in the Strand7 model due to the significance of its impact on the pretensioned concrete rib elements [7].

### 2.4. Podium

Analysis of Jorn Utzon's [8] Yellow and Red Book [9] designs of the structure shows that the apparent solid concrete podium of the Sydney Opera House is in fact a complex hollow structural network used to fit all the required facilities within the confined space of the unique structural design. The complexity of this subterranean design is beyond the scope of the analysis and hence is not included in the Strand7 design.

### 2.5. Foundation

The design of the Sydney Opera House's structural foundations embedded deep within the Hawkesbury Sandstone of the Sydney Harbour basin is not readily available to the public due to security concerns for the valued structure, and hence, the foundations are not part of the structural analysis. Likewise, the foundations of the concrete shells embedded in the podium are not specified and are assumed in the Strand7 model to be fully fixed.

### 2.6. Load Path

The unique design of the Opera House cleverly caters for the application of loading to ensure the longevity of the structure. The tensioned curvature of the structure's shell allows the transfer of basic structural dead loads from the ridge of the shell to the podium, as demonstrated in Figure 1b. Figure 1d demonstrates the way tension is applied through the concrete ribs at the top of the section. At the location of the small white holes, cables are focused in this region, as this location is anticipated to experience tension under loading when imagining the deflected shape. Wind loads are addressed by the rigidity of the structure brought about by the pretensioning in both the longitudinal and lateral directions. The importance of the longitudinal pretensioning for lateral loading is the resistance to overturning that it provides when the structure is experiencing strong westerly or easterly winds. Similar to the resistance to dead loads, the longitudinal pretensioning provides rigidity to the shell, allowing a load path down to the foundations in the podium. Figure 1c shows the large surface area exposed to the wind for the western wing. Figure 1a shows that each wing of the structure will provide some level of wind protection for the other wing from respective winds. A northern or southern wind will compress or tension adjacent ribs of the structure by providing lateral prestress, and with the semi-arches attached to precast concrete segments, the ribs are able to be held together, retaining the structure's shape. Figure 1d shows the design of the lateral prestressing cable layout, demonstrating the progressive anchor design, noting again the position of the anchor on the outside of the shell where tension is expected to take place. Details of the structural elements are given in the following sections and summarised in Table 2.

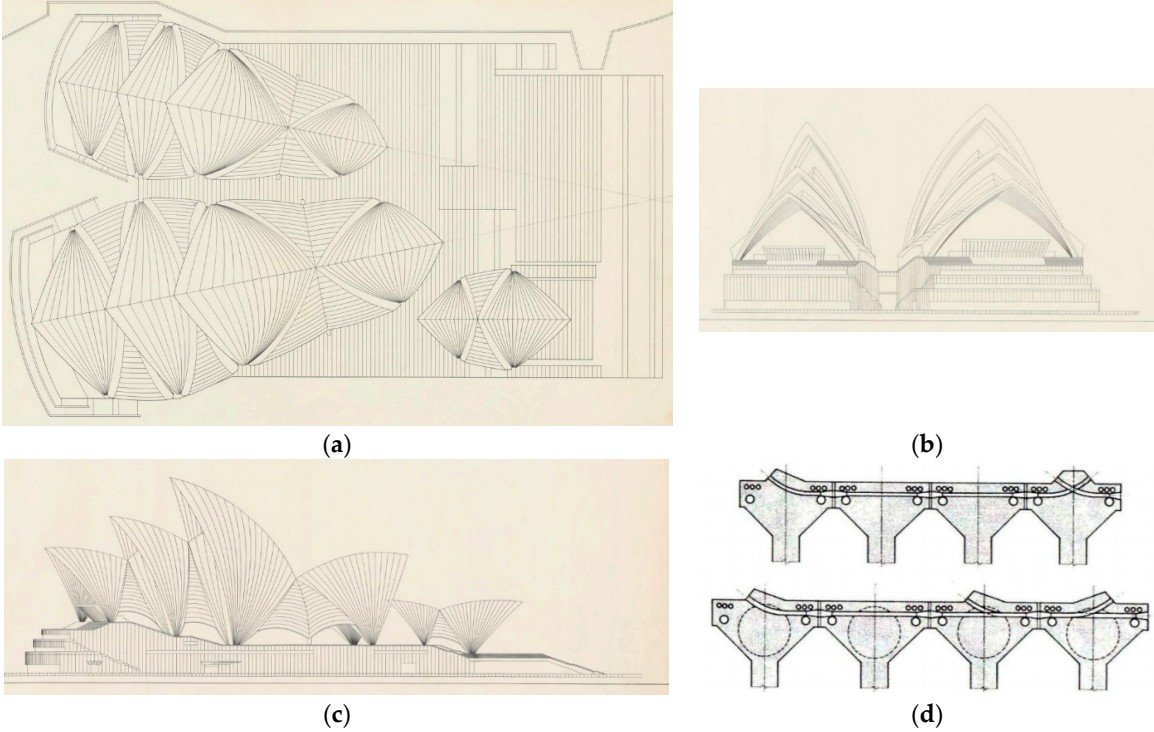

**Figure 1.** (**a**) Plan view, (**b**) north elevation, (**c**) west elevation (Utzon, 1962) Scale 1:32, (**d**) concrete rib cross-section ([8,9]).

**Table 2.** Structural elements of the Sydney Opera House.

| Details of the Structural Elements | Suggested Structural Element Sizes |
|---|---|
| Shells | All member sizes of the precast concrete shell structure were scaled from Jorn Utzon's Yellow Book details including elevations, curvature of interior geometry and thickness of the section, understanding that the external curvature follows the 75 m radius sphere concept. |
| Post-tensioning cables | 15.2 mm diameter strands (159 kN applied tension). (University of Sydney, 2016) |
| Semi-arches | Scaled from Yellow Book. |
| Precast concrete intermediate segment | Scaled from Yellow Book. |

Thermal loading for a structure such as the Sydney Opera House would generally be a significant issue given the close proximity of mass concrete and the potential for a large degree of thermal expansion. However, the Sydney Opera House designers innovatively incorporated the use of epoxy resin between the precast concrete rib sections, effectively bonding the sections to avoid short-term swelling and shrinkage from separating or shearing adjacent ribs [11]. Thermal analysis of the Sydney Opera House in the finite element model is thus not required, given that in the numerical modelling of a compatible structure, bricks connected do not separate or slip along their connection and therefore effectively replicate the actions of the epoxy as though it had perfect efficiency.

*2.7. Model Layout*

Creation of the finite element model of the Sydney Opera House required many assumptions and scaling of original structural design documents with reference to known data, such as height, to produce the shape of the structure [9]. Given the scope of the study as a preliminary analysis, these assumptions are valid for the work to be performed, and any results should be treated as preliminary in the absence of as-built data. Due to the symmetry of the Sydney Opera House, only half of the model was constructed and then mirrored for ease. As demonstrated earlier, the shells of the structure are all constructed as though from the one sphere of radius 75 m. Thus, in the creation of the Strand7 model, a node was created 75 m from the anticipated shell edges so that a user-defined coordinate system (USC) could be defined as a local spherical coordinate system with the node at its origin, allowing accurate modelling of the shell edge. Due to the variable thickness of the concrete ribs along the curvature of the structure, extrusion of elements could not be performed, and the internal curvature had to be approximated from design drawings [9]. The model consisted of two layers of Hexa16 elements making up the main body of the shell, Wedge15 elements constructing the intermediate precast concrete segments and Hexa20 elements at the connections of shells and precast segments allowing the compatibility of the model throughout the structure. Finally, truss beams were drawn longitudinally and laterally between each element of the two Hexa16 layers to mimic the post-tensioned steel. It is assumed that a single beam element applied to the model accounts for six tensioned strands longitudinally and four laterally. The material properties of the concrete and steel sections used in the model are given in Table 3.

**Table 3.** Strand7 input material properties.

| Material Properties | Strand Elements | Young's Modulus (MPa) | Density (kg/m$^3$) | Poisson's Ratio | Sectional Area (m$^2$) |
|---|---|---|---|---|---|
| Concrete (40 MPa) | Brick | 32,800 | 2400 | 0.2 | N/A |
| Steel | Longitudinal Beam | 200,000 | 7850 | 0.3 | 0.006567 |
| Steel | Lateral Beam | 200,000 | 7850 | 0.3 | 0.002919 |

## 3. Analysis under Dead and Wind Loads

### 3.1. Dead Loads

The structural dead load of the Sydney Opera House was calculated through the input of member sizes and material properties data into the Strand7 software, combined with the application of gravity loading (assuming 9.81 m/s$^2$ in the negative Y-direction, which is vertically downward). For the analysis of stress distribution in brick elements throughout this study, the 33 principal stress condition in Strand7 software is used. This analysis condition depicts the maximum and minimum stresses experienced within the plane of the structure's elements, demonstrating the stresses experienced in the concrete along the same plane as that for the prestressed cables under the axial stress condition. Figure 2a,b shows the maximum stress experienced in both the prestressed cables and concrete occurs at the base of the structure, while Figure 2c,d shows the vertical deflections to be greatest a quarter of the way up the shells, which aligns with ideas of flexing concave structures and the presence of rigid base connections.

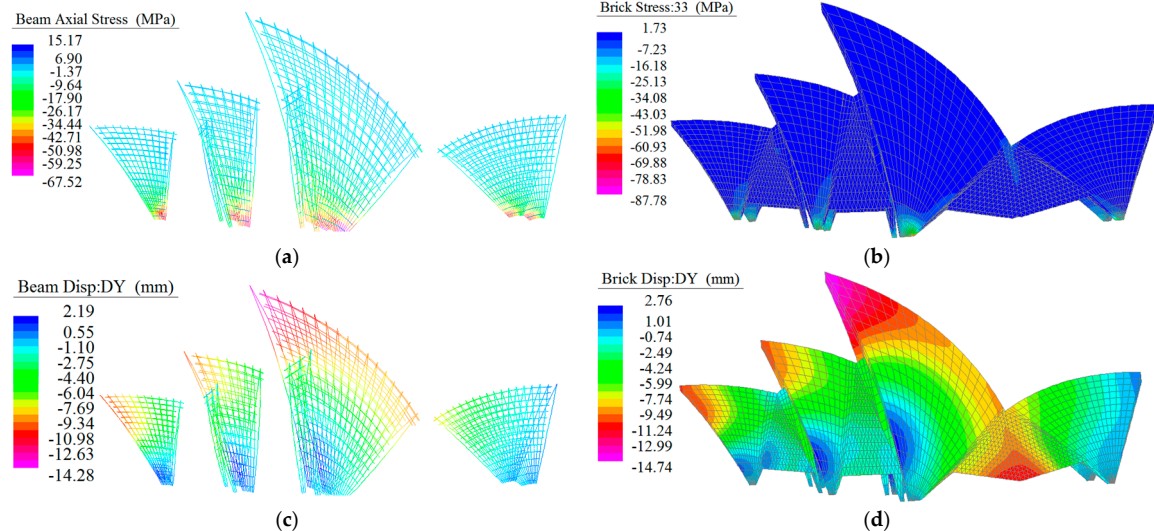

**Figure 2.** Strand7 dead load outputs (unscaled) (**a**) cable stress, (**b**) concrete stress, (**c**) cable displacement, (**d**) concrete displacement.

### 3.2. Wind Loads

Wind loads for the Sydney Opera House were calculated by developing the equivalent global face pressure experienced by the structure from the Australian Standards [12], in collaboration with information from [13] on the influence of the complex shape of the structure, for implementation into the Strand7 model, over the western face of the structure. The wind load equations are given in Equations (1) and (2) with the variables defined in Table 4.

**Table 4.** Five year probability of expected wind data calculations from [12].

| Variable | Result | Reasoning |
|----------|--------|-----------|
| $V_R$ | 32 m/s | (Region A2 : Sydney) |
| $M_{z,cat(h=0\text{m})}$ | 0.75 | (*where h* = 0 m) |
| $M_{z,cat(h=75\text{m})}$ | 0.98 | (*where h* = 75 m) |
| $M_{z,cat(h=65\text{m})}$ | 0.948 | (*where h* = 65 m) by linear interpolation |
| $V_{des,\theta(h=0\text{m})}$ | 24 m/s | $V_{des,\theta} = 32 \times 1(0.75 \times 1 \times 1)$ |
| $V_{des,\theta(h=65\text{m})}$ | 30.336 m/s | $V_{des,\theta} = 32 \times 1(0.948 \times 1 \times 1)$ |
| $F_{des(h=0\text{m})}$ | $(0.1658 \times A_z)$kPa | $(0.5 \times 1.225)[24]^2 \times 0.47 \times A_z$ |
| $F_{des(h=65\text{m})}$ | $(0.2649 \times A_z)$kPa | $(0.5 \times 1.225)[30.336]^2 \times 0.47 \times A_z$ |

$$V_{des,\theta} = V_R M_d (M_{z,cat} M_s M_t) \begin{cases} V_R : \text{ Regional gust wind speed for annual} \\ \qquad \text{probability of exceedance data} \\ M_d = 1 : \text{ Wind direction multiplier (West)} \\ M_{z,cat} : \text{ Terrain/Height multiplier (Category 4)} \\ M_s = 1 : \text{ Shielding multiplier} \\ M_t = 1 : \text{ Topographic multiplier} \end{cases} \tag{1}$$

$$F = (0.5\rho_{air})\left[V_{des,\theta}\right]^2 C_D A_z \begin{cases} \rho_{air} = 1.225 \text{ kg/m}^3 \\ V_{des,\theta} : \text{ Location design wind speed} \\ C_D = 0.47 : \text{ Drag coefficient of a Sphere (Hoerner, 1965)} \\ A_z : \text{ Area of exposed surface to wind pressure} \end{cases} \tag{2}$$

By varying the annual probability of exceedance for the intensity of the wind experienced at the site of the Sydney Opera House and interpolating the height data based on the Terrain/Height multiplier from AS/NZS 1170.2:2011 [12], wind pressures of varying intensities can be applied to Strand7 as an equation of the global face pressure up the western face of the model, acting in the negative x-direction, noting that the Area ($A_z$) is taken from the model, due to the complexity of the shape. For ease in modelling, the Terrain/Height multiplier is assumed to be linearly interpolated from the base of the structure to the tip, whereas it remains constant up to 30 m height before gradually increasing with altitude in the Australian Standards. This is an acceptable simplification as it is a contingent application of pressure greater than the standard case. The following demonstrates a five-year probability of exceedance wind case. Figure 3 below shows that the maximum stress induced by the wind loading takes place at the base of the structure, which is to be expected, confirming the need to ensure the capacity of the structure's connection to its foundations. Figure 3a shows the two sides of each structural shell experience opposite stresses to cater for the westerly wind loading: The eastern face is put into compression suppressing overturning forces, while the western face is in tension holding the shells together. Figure 3b also demonstrates that the brick elements of the structure almost entirely experience compression stresses from the wind loading, consistent with the principles of prestressing as the cables take up the tension forces from the load. The brick sections that experience tensile stresses from loading will, however, remain in compression throughout the concrete cross-section due to the action of the prestressed cables, therefore retaining the strength and shape of the structure. The maximum vertical displacement of the structure was on the edge of the shells at around mid-height, which is due to the section's free boundary condition at this location and the general understanding of deformation of a concave shell. Given that loading is a function of both height and area for wind application, the maximum load can be approximated with height, and this deformation demonstrates the need for stiffness of the section through the use of pretensioning.

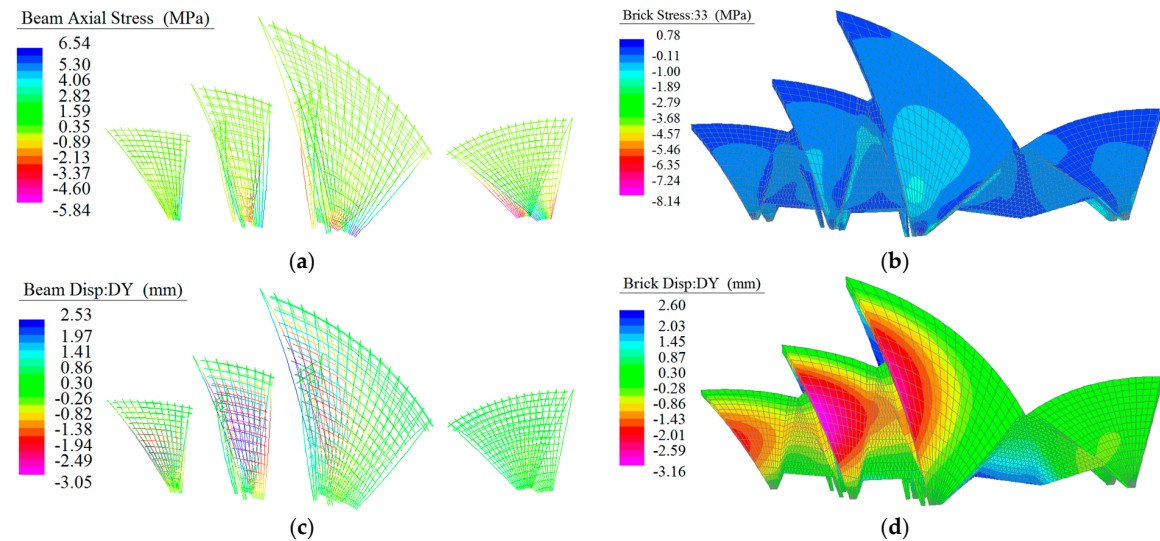

**Figure 3.** Strand7 five-year wind outputs (unscaled). (**a**) Cable stress, (**b**) concrete stress, (**c**) cable displacement, (**d**) concrete displacement.

By applying this methodology for wind intensities of up to 500 years probability of exceedance and inputting the data into Strand7, the following Y-displacements (vertically downward) for the prestressed cabling are recorded for comparison and are shown in Table 5.

**Table 5.** Impact of wind load on deflection of prestressed cabling.

| Case | Average $V_{des,\theta}$ (m/s) | Disp. Y (mm) |
|---|---|---|
| 5 year | 27.17 | 3.16 |
| 20 year | 31.41 | 11.0 |
| 50 year | 33.11 | 12.3 |
| 100 year | 34.81 | 13.5 |
| 500 year | 38.21 | 16.5 |

Table 5 shows that the deflection of the structure remains very small, within acceptability for all wind cases, reaching a worst case of 16.5 mm. It is assumed that these deflections lie within the elasticity range of the structure and therefore cause no long-term deflections. As such, the wind loading is excluded from the numerical analysis of creep and shrinkage, given that it is a transient load [14].

## 4. Analysis of Creep and Shrinkage

### 4.1. Model Inputs

With the model finalised, the next stage in the development of the analysis was the calculation of the required data inputs for the analysis of creep and shrinkage dictated by the input fields of the quasistatic solver. This solver is compatible with European and American design standards only, and hence, due to the similarities between Australian and European codes and the common use of the metric system, the required inputs were calculated using CEB-FIP Model Code 90 ([15–17]). The solvers used for each analysis type is given in Table 6.

**Table 6.** Strand7 solvers.

| Analysis | Solver |
|---|---|
| Dead and Wind Load | Linear Static |
| Creep and Shrinkage | QuasiStatic |

### 4.1.1. Creep Analysis

The formulation and input values for the creep analysis of the model are given in Equations (3)–(13). As a result of the changing cross-section of the concrete rib elements, a sample section, shown in Figure 4, was taken and assumed to be constant throughout the structure to simplify the concrete creep and shrinkage analysis calculations. In the absence of detailed geometric data for rib cross-sections, the sample cross-section was based on the building's concourse in [18] aligning with the sections detailed in [11]. This assumption is deemed to be contingent given the concourse's experience of heightened loading conditions, inclusive of the extensive live loads produced in the venue. Because this analysis is a preliminary approximation of the creep and shrinkage experienced in the Sydney Opera House to gauge the requirement for further analysis, these simplifications will not greatly affect the study. The resulting required inputs for Strand7 calculations of creep are shown in Table 7.

$$\phi(t, t_0) = \phi_0 \beta_c (t - t_0) \begin{cases} \text{Clause 2.1.6.4.3 CEB-FIP Model Code 90, where :} \\ \phi_0 : \text{ Notional creep coefficient} \\ \beta_c (t - t_0) : \text{ Creep development in time after loading} \\ t : \text{ age of concrete in days} \\ t_0 : \text{ age of concrete at first loading in days} \end{cases} \tag{3}$$

$$\phi_0 = \phi_{RH} \beta(f_{cm}) \beta(t_0) \begin{cases} \phi_{RH} : \text{Relative Humidity co-efficient} \\ \beta(f_{cm}) : \text{parameter for concrete compressive strength} \\ \text{at age 28 days} \\ \beta(t_0) : \text{ parameter based on the age of} \\ \text{concrete at first loading} \end{cases} \tag{4}$$

$$\beta_c(t - t_0) = \left[ \frac{\frac{t - t_0}{t_1}}{\beta_H + \frac{t - t_0}{t_1}} \right]^{0.3} \begin{cases} t_1 = 1 \text{ day} \\ \beta_H : \text{a parameter based on humidity and member size} \end{cases} \tag{5}$$

$$\phi(t, t_0) = \phi_{RH} \beta(f_{cm}) \beta(t_0) \left[ \frac{\frac{t - t_0}{t_1}}{\beta_H + \frac{t - t_0}{t_1}} \right]^{0.3} \text{Substituting (2) and (3) into (1)} \tag{6}$$

$$\phi(t, \tau) = \left( \frac{(t - \tau)^\alpha}{\beta + (t - \tau)^\alpha} \right)^\delta \phi_u \begin{cases} \text{Strand7 expression for the Hyperbolic Law} \\ \tau : \text{ age of concrete at first loading} \\ t : \text{ current age of concrete} \end{cases} \tag{7}$$

$\alpha = 1.0$
$\beta = \beta_H$ Equating (4) and (5) the parameters for Strand7 inputs are found
$\delta = 0.3$

$$\phi_u = \phi_{RH} \beta(f_{cm}) \beta(t_0) \text{ Likewise equating (4) and (5)} \tag{8}$$

$$\beta_H = 150 \left\{ 1 + \left( 1.2 \frac{RH}{RH_0} \right)^{18} \right\} \frac{h}{h_0} + 250 \le 1500 \begin{cases} \text{RH is the Relative Humidity of the ambient} \\ \text{environment, assumed to be 80\% for the} \\ \text{outdoor marine conditions of the harbour} \\ RH_0 = 100\% \\ h_0 = 100 \text{ mm} \end{cases} \tag{9}$$

$$h = \frac{2A_c}{u} \begin{cases} h : \text{Notational size of the member in mm} \\ A_c : \text{Member cross section} \\ u : \text{ perimeter of the member in contact} \\ \text{with the atmosphere} \end{cases} \tag{10}$$

$$\beta(f_{cm}) = \frac{5.3}{\left(\frac{f_{cm}}{f_{cmo}}\right)^{0.5}} = \frac{5.3}{\left(\frac{40}{10}\right)^{0.5}} = 2.65 \begin{cases} f_{cm} : \text{mean compressive strength} \\ \text{of concrete at age 28 days (40 MPa)} \\ f_{cm0} : \text{mean compressive strength} \\ \text{of concrete at age 7 days (10 MPa)} \end{cases} \tag{11}$$

$$\beta(t_0) = \frac{1}{0.1 + \left(\frac{t_0}{t_1}\right)^{0.2}} \quad \begin{array}{l} \text{In Strand7, the dependence of the parameter } \phi_u \text{ on} \\ \text{the age of first loading can be accounted by using a} \\ \text{factor vs time table. The parameter } \beta(t_0) \text{ can be} \\ \text{defined as a factor, so that the parameter } \phi_u \text{ can} \\ \text{be defined as } \phi_u = \phi_{RH}\beta(f_{cm}). \end{array} \tag{12}$$

$$\phi_u = \phi_{RH}\beta(f_{cm}) = 1.285 \times 2.65 = 3.4 \tag{13}$$

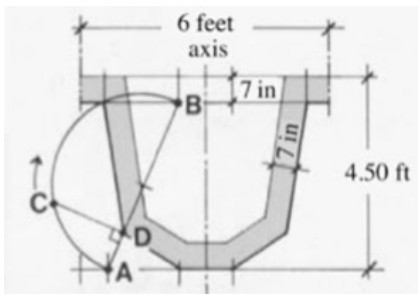

$$Ac = \sqrt{(4.5 \times 304.8)^2 + (3 \times 304.8)^2} \times 7 \times 25.4 \times 2$$
$$= 586191\text{mm}^2$$

$$u = \sqrt{(4.5 \times 304.8)^2 + (3 \times 304.8)^2} \times 2 = 3297\text{mm}$$

$$\therefore \text{h} = \frac{2A_c}{u} = 2 \times \frac{586191}{3297} = 355\text{mm}$$

$$\therefore \beta_H = 150\left\{1 + \left(1.2\frac{80}{100}\right)^{18}\right\}\frac{355}{100} + 250 = 1037$$

**Figure 4.** Sample concrete cross-section (**left**) [18] unscaled, calculations for sample concrete section (**right**), conversion factors 1ft = 304.8 mm, 1in = 25.4 mm.

**Table 7.** Strand7 creep input values.

| | |
|---|---|
| $\alpha$ | 1.0 |
| $\beta$ | 1037 |
| $\delta$ | 0.3 |
| $\phi_u$ | 3.4 |

### 4.1.2. Shrinkage Analysis

The formulation and input values for the shrinkage analysis of the model are given in Equations (14)–(22). Details on the cement type are given in Table 8, and the required inputs for Strand7 calculations of shrinkage are shown in Table 9.

$$\varepsilon_s(t, t_s) = \varepsilon_{cso}\beta_s(t - t_s) \begin{cases} \text{Clause 2.1.6.4.4 CEB-FIP Model Code 90, where :} \\ \quad \varepsilon_{cso} : \text{ Nominal shrinkage coefficient} \\ \beta_s(t - t_s) : \text{Shrinkage development in time after loading} \\ \quad t : \text{ age of concrete in days} \\ \quad t_s : \text{ age of concrete at first loading in days} \end{cases} \tag{14}$$

$$\varepsilon_{cso} = \varepsilon_s(f_{cm})\beta_{RH} \begin{cases} \varepsilon_s(f_{cm}) : \text{parameter for concrete mean compressive} \\ \quad \text{strength at age 28 days} \\ \beta_{RH} : \text{ shrinkage coefficient accounting} \\ \quad \text{for relative humidity} \end{cases} \tag{15}$$

$$\beta_s(t - t_s) = \left[ \frac{\frac{t-t_s}{t_1}}{350\left(\frac{h}{h_0}\right)^2 + \frac{t-t_s}{t_1}} \right]^{0.5} \left\{ \begin{array}{l} h : \text{notional size of the member in mm} \\ \qquad h_0 : 100 \text{ mm} \\ \qquad t_1 : 1 \text{ day} \end{array} \right. \tag{16}$$

$$\varepsilon_s(t, t_s) = \varepsilon_s(f_{cm})\beta_{RH} \left[ \frac{\frac{t-t_s}{t_1}}{350\left(\frac{h}{h_0}\right)^2 + \frac{t-t_s}{t_1}} \right]^{0.5} \text{Substitute (12) and (13) into (11)} \tag{17}$$

$$\varepsilon_s = \left( \frac{(t - \tau)^{\alpha_s}}{\beta_s + (t - \tau)^{\alpha_s}} \right)^{\delta_s} \varepsilon_{s0} \left\{ \begin{array}{l} \text{Strand7 expression for the Hyperbolic Law} \\ \quad \tau : \text{ age of concrete at first loading} \\ \quad t : \text{ current age of concrete} \end{array} \right. \tag{18}$$

$$\begin{array}{l} \alpha_s = 1.0 \\ \delta_s = 0.5 \end{array} \text{Equating (14) and (15)the parameters for Strand7 inputs are found}$$

$$\beta_s = 350\left(\frac{h}{h_0}\right)^2 = 350\left(\frac{355}{100}\right)^2 = 4411 \text{ Likewise equating (4) and (5)} \tag{19}$$

$$\varepsilon_{s0} = \varepsilon_s(f_{cm})\beta_{RH} \text{ Likewise equating (4) and (5)} \tag{20}$$

$$\varepsilon_s(f_{cm}) = \left[ 160 + 10\beta_{sc}\left( 9 - \frac{f_{cm}}{f_{cmo}} \right) \right] \times 10^{-6} \tag{21}$$

$\beta_{sc}$ : coefficient based on cement type from Table 8 below.

**Table 8.** Cement type for $\beta_{sc}$ from Clause 2.1.6.4.4 CEB-FIP Model Code 90 [17].

| Type of Cement | Slowly Hardening Cements SL | Normal or Rapid Hardening Cements N and R | Rapid Hardening High Strength Cements RS |
|---|---|---|---|
| $\beta_{sc}$ | 4 | 5 | 8 |

**Table 9.** Strand7 shrinkage input values.

| | |
|---|---|
| $\alpha_s$ | 1.0 |
| $\beta_s$ | 4411 |
| $\delta_s$ | 0.5 |
| $\varepsilon_{s0}$ | 0.00027328 |

Due to the abnormal shape of the Sydney Opera House, it is assumed that a rapid hardening high strength cement was used to set the shape of the precast concrete segments.

$$\therefore \varepsilon_s(f_{cm}) = \left[ 160 + 10 \times 8\left( 9 - \frac{40}{10} \right) \right] \times 10^{-6} = 0.00056$$

$$\beta_{RH} = 1 - \left( \frac{RH}{RH_0} \right)^3 = 1 - \left( \frac{80}{100} \right)^3 = 0.488 \left\{ \begin{array}{l} RH \text{ is the Relative Humidity of the ambient} \\ \text{environment, assumed to be 80\% for the} \\ \text{outdoor marine conditions of the harbour} \\ \qquad RH_0 = 100\% \end{array} \right. \tag{22}$$

*4.2. Obtained Results*

4.2.1. Creep Effect

By applying the input variables in the Strand7 software, the following data are retrieved for creep. Figure 5 shows the creep factor over time, over 100 years, and shows that the majority of creep effects occur in the first 30 years of prestress loading.

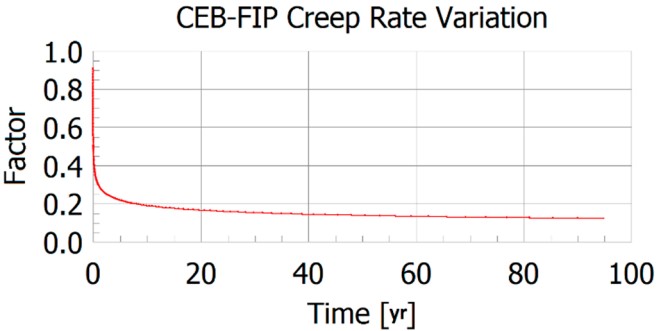

**Figure 5.** Creep factor versus time plot.

Figure 6 demonstrates the vertical creep strain due to dead load experienced throughout the structure for use in calculation of the prestress losses, noting that maximum strain occurs at the base of the structure due to the dependency of creep on loading.

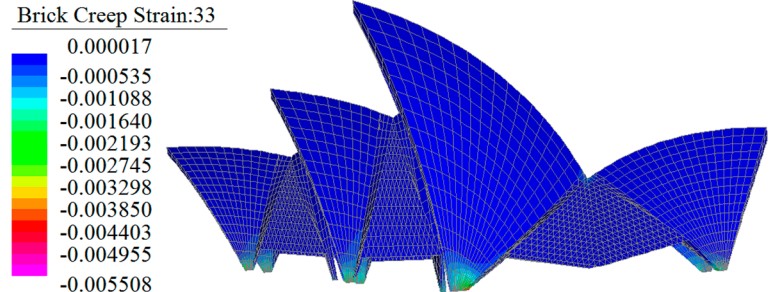

**Figure 6.** Strand7 creep strain contour (unscaled).

Finally, Figure 7 demonstrates the resultant stresses and displacements of the structure in the presence of creep to understand the subsequent impacts of creep on the concrete structure.

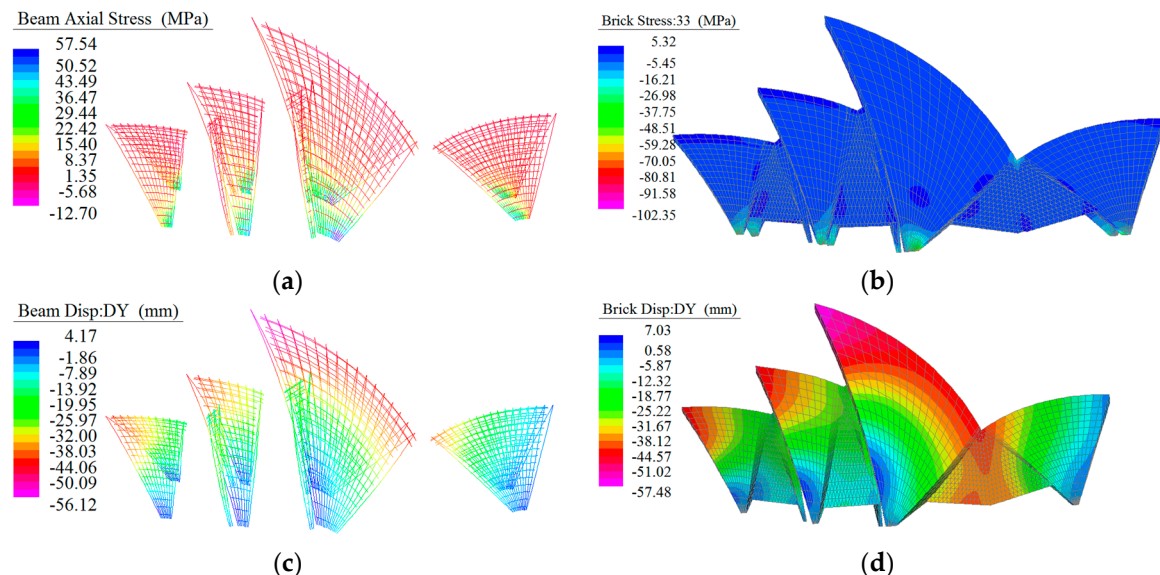

**Figure 7.** Strand7 creep outputs (unscaled) (**a**) cable stress, (**b**) concrete stress, (**c**) cable displacement, (**d**) concrete displacement.

### 4.2.2. Shrinkage Effect

By applying the input variables in the Strand7 software, the following data are retrieved for shrinkage. Figure 8 shows the shrinkage factor over time, over 100 years. Similar to the creep factor, the majority of shrinkage takes place within the first 30 years of prestress loading.

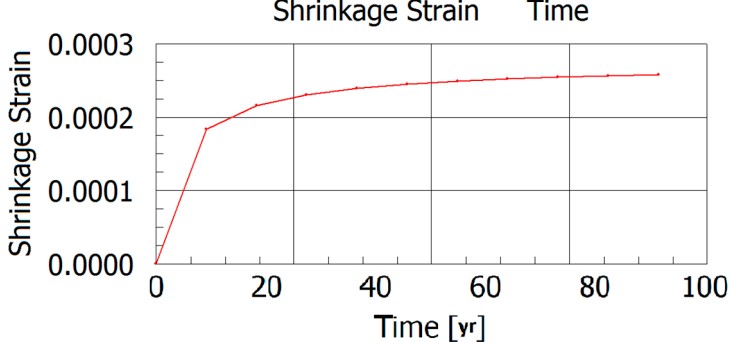

**Figure 8.** Shrinkage Strain Versus Time.

Figure 9 demonstrates the shrinkage strain experienced throughout the structure, which is uniform, given that it is dependent on factors of time, concrete strength and humidity, which remain constant over the entire structure. The resulting impacts of shrinkage such as stress and displacement are expected to be uniform throughout the structure.

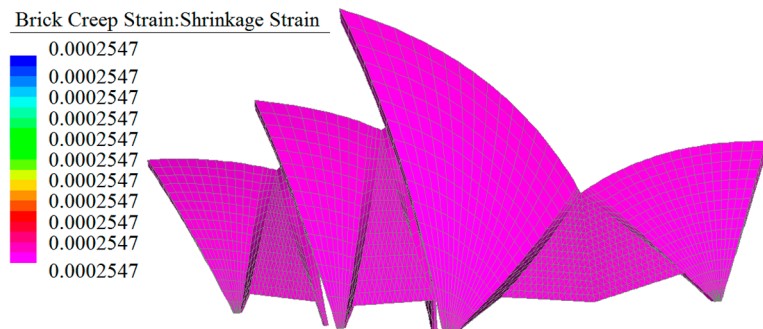

**Figure 9.** Strand7 shrinkage strain contour (unscaled).

### 4.2.3. Mesh Quality

Computational error is present around the base of the structure at the fixed connection to the podium, due to the elongation of the mesh elements brought about by the complex shape of the structure. This error may be reduced by application of subdivision of the mesh. However, this task is difficult and very intricate in the complex 3D model to ensure compatibility of the mesh, given that each section of elements for each structural component must be subdivided appropriately to align with the connecting elements. For example, subdivision of Hexa16 elements will result in a need to subdivide the Hexa20 elements attached to maintain mesh compatibility, and subsequently, the Wedge15 elements will also require adjustment. Further, subdividing to cater for the elongated base of the structure would result in elongation of other sections of the model, similar to the elongation present in the edges of the structure where subdivision would result in elongated surfaces of the structural model. Domain approximation error is extensive in this model given the lack of information for the construction of the model. Sections such as the interior curvature of the shells and the shape of the semi-arches had to be approximated and scaled from structural design drawings [9], resulting in this error compared to the as-built design case. However, given the use of isometric elements to approximate the structure's curved shape and the preliminary nature of this study, this is acceptable, and error is minimised.

### 4.3. Discussion of Creep and Shrinkage Analysis

The total stress losses experienced in the pretensioned cables of the Sydney Opera House result from strains in concrete ribs undergoing axial shortening from long-term creep and shrinkage, which in turn induces strain in the steel due to the compatibility of the steel–concrete connection, demonstrated in the Strand7 contour outputs in Figures 6 and 9. These stress losses follow the elastic modulus equation as shown in Equation (23):

$$\sigma_{Stress\ loss} = E \times \Delta\varepsilon_{steel} \tag{23}$$

where the strain of the steel matches that of the concrete due to the bonding required under Australian Standards (Equation (24)).

$$\therefore \Delta\varepsilon_{steel} = \Delta\varepsilon_{Concrete} = \varepsilon_{Creep} + \varepsilon_{Shrinkage} \tag{24}$$

Table 10 below demonstrates the Strand7 data outputs for the creep and shrinkage impacts that have taken place in the Sydney Opera House from the time of pretension loading to the present and on to 2050. Unlike shrinkage, creep is not constant throughout the structure's cross-section due to its load dependency and hence must be averaged throughout the critical cross-section, taken to be the concrete rib and strands of the tallest section of the structure, to ensure a correct time-dependent analysis. The creep strain is concentrated at the base of the concrete rib and reduces with height up the section. Stress loss has been calculated from the elastic modulus equation above, and the stress per strand is calculated by multiplication of the stress loss (MPa) by the cross-sectional area of a strand ($A_{strand} = \pi \times \frac{15.2^2}{4} = 181.46$ mm$^2$).

**Table 10.** Impact of wind load on deflection of prestressed cabling.

| Parameter | Present | 2050 |
|---|---|---|
| $\varepsilon_{shrinkage}$ | 0.000245 | 0.000255 |
| avg. $\varepsilon_{creep}$ | 0.00064 | 0.000643 |
| Total Strain (%) | 0.0885 | 0.0898 |
| $\sigma_{stress\ loss}$ (MPa) | 177.0 | 179.6 |
| $F_{loss\ per\ strand}$ (kN) | 32.12 | 32.59 |

The implication of stress loss in pretensioned concrete has great consequences, as stress loss will reduce the compression effect of the concrete in the structure, potentially leaving areas of the concrete cross-section in tension where it is vulnerable. At the time of its completion in 1973, Arup engineers published their comprehensive analysis of the structural forces and loading expected to impact the Sydney Opera House, expressing specifically that creep and shrinkage of concrete had been catered for [11]. They had anticipated a shrinkage strain of 0.03%, a conservative assumption in comparison to the results of the current study, while also expecting an ultimate loss in pretensioning of cables at 33%, with 5% owing to creep and shrinkage. With an approximate 20% prestress loss resulting from creep and shrinkage, the results of this study appear to be dire. However, the analysis by Arup was quoted to approximately 50 years, whereas this study describes a much broader time period, from initial tensioning to the year 2050, inferring that the results remain valid and acceptable [11].

The important message from the data presented in this study is that the majority of creep and shrinkage impacts have occurred in the first 30 years of pretensioned loading and hence, at the present time of about 50 years after construction, the major structural concerns of the Sydney Opera House have been surpassed. Only a further 0.47 kN of prestress loss is anticipated to occur between now and 2050 in each strand within the structure. Given that cables were initially loaded with 159 kN of tension this would not be enough to cause structural damage to the Sydney Opera House. However, it may cause slight surface tension cracking of the concrete, which would allow water ingress. The conservation

protocol suggested includes regular inspection of the concrete, described by the University of Sydney research into automated systems, in collaboration with the use of Engineered Cementitious Composites, using the composites ductility to seal surface abrasions, protect against corrosion of embedded steel and provide a strain hardening effect on the existing concrete [19–21].

## 5. Suggestions for Further Studies

Research into the methods of managing structural issues related to creep and shrinkage had been performed in anticipation of major damage. Three methods for recovery of stress loss were examined to determine their effectiveness and can be replicated to other historical buildings. Firstly, the idea of simply retensioning the cables to their original state was investigated. However, the cables are bonded to the concrete via grout, as bonded pretensioning is required under Australian Standards for safety reasons, meaning that tensioning of the cables will also tension the concrete and cause further structural problems. Secondly, carbon fibre strapping solutions were studied to show the benefits of an application of uniform compression about the concrete section in stiffening the element. However, due to the complete nature of the shell section, intrusion of the concrete would be required to implement the solution, causing damage that would outweigh the resulting benefits [22–24]. Finally, installation of new pretensioned cabling was suggested by drilling through the concrete and pulling through new cabling [25–28] However, the intrusion of the concrete to implement the solution poses the major risk of further damaging the Sydney Opera House structurally before implementation of the new prestressing.

Structural damage may occur by providing localised high stress in a prestressed section by forming drill holes. Further, delicate studies of the way new prestressing impacts on the old prestressing would also be required given the eccentric loading conditions the solution would provide. Therefore, the Sydney Opera House would require a far more detailed and purpose-built solution to any structural creep and shrinkage issues if they occur. Figure 10 illustrates the recent situation of the Opera House.

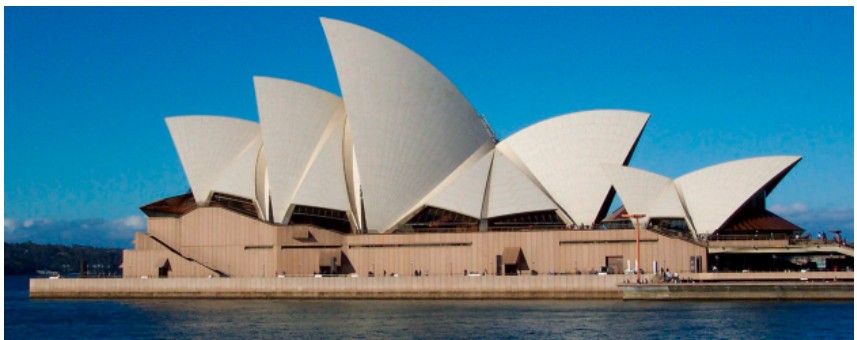

**Figure 10.** Recent photo of the Opera House.

As mentioned, a building information model (BIM) of the Sydney Opera house was created based on as-built information for the management of on-going maintenance works on the site. The current practices of as-built modelling suggest using different types of laser scanners to collect accurate data from indoor and outdoor areas [29]. The BIM aimed to revolutionise the facilities management of the Opera House by sharing the information with all stakeholders on the area of the structure requiring works and ongoing structural assessment ([3,4]). However, the technology is advancing quickly, and there are still more useful technologies that can help the facility managers to carefully monitor similar historical buildings in the future. In fact, future studies can focus on the utilisation of a digital model and a digital twin for monitoring and documenting heritage buildings. A digital twin can be developed based on a CAD model linked to sensors installed on a matching physical object. Then, the digital twin will be used in building lifecycle management for monitoring the building remotely in real time. In order to facilitate the uptake of these technologies, future studies should develop new applications for monitoring buildings. The technology adoption closely depends on the

compatibility and interoperability of numerical analysis programs with building information modelling, usefulness of the new technologies, vendor support, and applicability of these technologies [30–39].

## 6. Conclusions

This paper provides a preliminary quasistatic analysis of the effects of creep and shrinkage on the Sydney Opera House under the conditions of structural dead load, to activate the potential use of finite element software in collaboration with complex building information models. The Strand7 results established that over the last 50 years, the structure has experienced 32.12 kN of prestress losses in each strand of steel embedded in the pretensioned concrete. Predictions of the structure's future to 2050 show an additional anticipated loss of only 0.47 kN in prestressing, demonstrating the longevity of the Sydney Opera House's structure and the change in conservation focus from structural concerns to inspection and maintenance of minor issues such as surface cracking and water ingress. The integration of building information modelling and finite element analysis would allow the creation of a more accurate model with as-built data to provide an in-depth analysis of the creep and shrinkage of the entire structure, inclusive of the podium and subterranean floors of the Sydney Opera House's structure.

**Author Contributions:** Conceptualisation: All authors; Literature review: All authors; Research Method: All authors; Software: D.F., L.O.W., Z.L. and K.Z. Drafting the article based on the numerical reports: F.T. and S.S.; Resources and interpretation: F.A.M. and F.T.

**Funding:** This research received no external funding.

**Acknowledgments:** Authors would like to express deepest appreciation to the University of New South Wales and the University of Sydney for providing convenient places to undertake the current research.

**Conflicts of Interest:** The authors declare no conflict of interest.

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
