# Peer review of "Numerical Analysis of the Creep and Shrinkage Experienced in the Sydney Opera House and the Rise of Digital Twin as Future Monitoring Technology"

_buildings, doi:10.3390/buildings9060137_

Reviewer 1 Report

The subject matter of the paper is within the scope of the journal and has a good technical quality. The article draws attention to the phenomenons of creep and shrinkage of concrete occuring in the Sydney Opera House. To solve the scientific problem numerical analysis were used. The authors carried out a wonderful work, however there are some observations and comments raised which the authors needs to address or correct:

(1) For the enrichment of the article it would be good to show a current photo of the Opera House.

(2) Please specify how the important parameter t0 was determined. After so many years since putting the construction into service certainly it was  not easy. Are there any historical data on this topic?

(3) How can the salinity from the ocean affect the rheology and the weakening of concrete in the  Sydney Opera House structure. Please discuss this particular problem.

(4) Significant impact on the rheological parameters of the concrete, mainly shrinkage, has the type of cement used. Does the assumption that rapid hardening high strength cement has been used to perform the Sydney Opera has some confirmation in the preserved projects, or is it just an assumption of the authors.

(5) The description of the used numerical procedure (Section 4.2.3) are not sufficiently presented and only basic results of simulations are mentioned. For example, it is required to summarizes information about types of elements and numbers of nodes used in numerical calculations.

(6) The article presents an analysis of the assessment of concrete wear in a structure based on 2 rheological parameters. The full analysis should also include an assessment of the structure of the material in the light of the occurring microcracks. Are such tests planned?

In this field, I think, the authors must strengthen the References section with the references that use this experimental technique. It is therefore required to discuss and cite these papers:

“An assessment of microcracks in the Interfacial Transition Zone of durable concrete composites with fly ash additives”, Composite Structures, 2018.

“Evaluation of morphology and size of cracks of the Interfacial Transition Zone (ITZ) in concrete containing fly ash (FA)”, Journal of Hazardous Materials, 2018.

Author Response

Reviewer’s   comments   ( First Reviewer )

Authors’ reply  

(1) For the enrichment of the article it   would be good to show a current photo of the Opera House.

A   recent photo of the Opera House was added to the article.

(2) Please specify how the important parameter t0   was determined. After so many years since putting the construction into   service certainly it was not easy. Are there any historical data on this   topic?

Of   course,  t0 is one of the   significant factors. However, determination of the  t0  is very complicated. Also, we did not have   access to the historical data. In the current analysis , an     arbitrary   number was assumed as  t0.

(3)   How can the salinity from the ocean affect the rheology and the weakening of   concrete in the Sydney Opera House structure? Please discuss this   particular problem.

This   is a unique question. Definitely, the salinity from the ocean can crucially   influence on the time deponent analysis as well as deformation of the casted   concrete in the Opera House. This is an individual problem and it needs to   allocate a spate research paper to that topic. At the moment, the evaluation   the effect of the salinity from the ocean is out of the scope of the current   paper. However, as indicated, this is very interesting and important topic.   This topic can be suggested for future research. 

(4)   Significant impact on the rheological parameters of the concrete, mainly   shrinkage, has the type of cement used. Does the assumption that rapid   hardening high strength cement has been used to perform the Sydney Opera has   some confirmation in the preserved projects, or is it just an assumption of   the authors.

Having   access to the some key technical parameters form Opera House is currently a   significant patent. 

Refer   to the current link,  https://sydney.edu.au/news-opinion/news/2016/09/06/conserving-a-concrete-masterpiece.html,   it is almost impossible or should be very expensive to request for collecting   some data which are relevant to the major parameters in creep and shrinkage. At   the moment, some reasonable assumptions by authors were assumed to   comprehensively undertake the simulations.   

(5)   The description of the used numerical procedure (Section 4.2.3) are not   sufficiently presented and only basic results of simulations are mentioned.   For example, it is required to summarizes information about types of elements   and numbers of nodes used in numerical calculations.

I would like   to appreciate for the suggestion. A further explanations regarding to how to   develop the finite element models as well as defining some initial input   parameter were added to the article.

(6)   The article presents an analysis of the assessment of concrete wear in a   structure based on 2 rheological parameters. The full analysis should also   include an assessment of the structure of the material in the light of the   occurring microcracks. Are such tests planned?

In   this field, I think, the authors must strengthen the References section with   the references that use this experimental technique. It is therefore required   to discuss and cite these papers:

“An   assessment of microcracks in the Interfacial Transition Zone of durable   concrete composites with fly ash additives”, Composite Structures, 2018.

“Evaluation   of morphology and size of cracks of the Interfacial Transition Zone (ITZ) in   concrete containing fly ash (FA)”, Journal of Hazardous Materials, 2018.

 The suggested references were added to the article.

Currently,   there is a substantial research plan is under way in the Opera House

(  https://pacetoday.com.au/robots-sensors-sydney-opera-house/),   which is very capital intensive.

Professor Gianluca Ranzi, University of Sydney said: “The project has provided the next   generation of engineers with a real-world opportunity to develop their skills   and increase their understanding of heritage-building conservation.“The use   of advanced technology has provided the basis for the development and   prototyping of an effective inspection strategy applicable to twentieth   century concrete buildings.” Located in a harsh marine environment the Opera   House requires vigilant maintenance and active conservation practices to   ensure its beauty and integrity are safeguarded. Three primary areas of the   Opera House concrete were studied, based on their heritage and structural   significance: the Sails structure; the Roof Pedestals; and the Northern   Broadwalk under-structure

However, the suggested references were added to the article.

Reviewer 2 Report

This paper needs some revisions, which should be considered:

Introduction is incomplete. It is required to add some references to show which types of shrinkage creates some problems and in the research topics what solutions have been proposed for concrete. Therefore, its recommended to use the following references as simple solutions to control shrinkage in the buildings and what the fundamental concept of shrinkage is:

https://www.sciencedirect.com/science/article/pii/S0950061818323043

https://ascelibrary.org/doi/abs/10.1061/(ASCE)MT.1943-5533.0001918

In Table 3, it is required to report elastic modulus

It is required to explain more about the constitutive model used for modeling of materials. This reference (https://www.sciencedirect.com/science/article/pii/S026382231630695X) could be useful for understanding the concept and modeling of the shells.

In figures 2 and 3, it is recommended to use Von Mises stresses.

Author Response

Reviewer’s   comments    second   Reviewer’s

Authors’ reply  

Introduction is incomplete

A further   explanation and clarification was added. Individually, a separate statement   was indicated why this research is significant in the introduction. 

It   is required to add some references to show which types of shrinkage creates   some problems and in the research topics what solutions have been proposed   for concrete. Therefore, its recommended to use the following references as   simple solutions to control shrinkage in the buildings and what the   fundamental concept of shrinkage is:

https://www.sciencedirect.com/science/article/pii/S0950061818323043

https://ascelibrary.org/doi/abs/10.1061/(ASCE)MT.1943-5533.0001918

https://link.springer.com/article/10.1007/s12205-017-1714-3

https://www.icevirtuallibrary.com/doi/abs/10.1680/jcoma.16.00077

I would like   to thank about the suggested references. The relevant references were   considered.

In   Table 3, it is required to report elastic modulus

The   elastic modulus was already provided.

It   is required to explain more about the constitutive model used for modelling   of materials. This reference   (https://www.sciencedirect.com/science/article/pii/S026382231630695X) could be useful for understanding the concept and   modelling of the shells. 

I would like   to thank about the suggested references. The relevant references were   considered.

In figures 2 and 3, it is recommended to use Von Mises stresses.

The   presented results are based on the major principal stresses ( or governed stresses), Von Mises stresses is based on the combination of   the principal stresses, thus, an individual stress at a particular govern   direction  can be more precise rather   than using a combination of the stresses.     

Reviewer 3 Report

1.The motivation of doing this numerical simulation is not clear in this paper. 2. How the authors decided dead load? 3. Mesh quality section need reference for mesh selection and its affect on final results 4. Authors can propose some future work using BIM analysis. It will improve the accuracy of obtain results. 5. Some typos mistakes are there. in section 4.

Author Response

Reviewer’s comments   (The Third Reviewer )

Authors’ reply

The   motivation of doing this numerical simulation is not clear in this paper.

The main   motivation of the current research is to determine a cost-effective method to   evaluate the time-dependent deformation of the structural elements in the   Opera House. Currently, there is a substantial research plan is under way in   the Opera House

(  https://pacetoday.com.au/robots-sensors-sydney-opera-house/),   which is very capital intensive. The current numerical method can be extended   to have a cost-effective replaced solution against the current field   measurement project which is very expensive.

2. How the   authors decided dead load?

The dead load is the calculated   self-weight of the structural and non-structural elements by the Finite   Element Package (STRAND 7).

3. Mesh quality   section need reference for mesh selection and its affect on final results.

The relevant explanation about the   mesh quality was broadly discussed.

4. Authors can propose some future work using BIM   analysis. It will improve the accuracy of obtain results.

The current research is relevant to   maintaining the icon/heritage building. Thus, it is relevant to structural   protect a historical building, thus, this is not relevant to the future   building as well as the future design that we could take into account the BIM   analysis. 

Sydney Opera House Director, Building Greg McTaggart said: “The   complexity of the Sydney Opera House structure called for a creative response   to its conservation. The timing of the Getty grant aligned with the   near-completion of the Opera House’s Conservation Management Plan (4th   Edition) and advances in our Building Information Management system. What is   unique and innovative about this project is that we will be able to   seamlessly integrate heritage policies with the day-to-day management of the   building fabric.”

5. Some typos   mistakes are there. in section 4.

I would like appreciate for the   comment. The section 4 was revised.

Round  2

Reviewer 1 Report

The required corrections have been made. Thus, I suggest to accept the manuscript in its current form

Reviewer 2 Report

Most comments were considered in the revised version.